# Analyzing native membrane protein assembly in nanodiscs by combined non-covalent mass spectrometry and synthetic biology

**Erik Henrich[1†], Oliver Peetz[2†], Christopher Hein[1], Aisha Laguerre[1], Beate Hoffmann[1], Jan Hoffmann[2], Volker Dötsch[1], Frank Bernhard[1], Nina Morgner[2]\***

[1]Institute of Biophysical Chemistry, Centre for Biomolecular Magnetic Resonance, J W Goethe-University, Frankfurt am Main, Germany; [2]Institute of Physical and Theoretical Chemistry, J W Goethe-University, Frankfurt am Main, Germany

**Abstract** Membrane proteins frequently assemble into higher order homo- or hetero-oligomers within their natural lipid environment. This complex formation can modulate their folding, activity as well as substrate selectivity. Non-disruptive methods avoiding critical steps, such as membrane disintegration, transfer into artificial environments or chemical modifications are therefore essential to analyze molecular mechanisms of native membrane protein assemblies. The combination of cell-free synthetic biology, nanodisc-technology and non-covalent mass spectrometry provides excellent synergies for the analysis of membrane protein oligomerization within defined membranes. We exemplify our strategy by oligomeric state characterization of various membrane proteins including ion channels, transporters and membrane-integrated enzymes assembling up to hexameric complexes. We further indicate a lipid-dependent dimer formation of MraY translocase correlating with the enzymatic activity. The detergent-free synthesis of membrane protein/nanodisc samples and the analysis by LILBID mass spectrometry provide a versatile platform for the analysis of membrane proteins in a native environment.

**\*For correspondence:** morgner@chemie.uni-frankfurt.de

[†]These authors contributed equally to this work

## Introduction

Crucial cellular processes depend on membrane proteins and their interactions. The majority of membrane proteins is organized in higher order complexes and oligomeric assembly is frequently used to modulate protein function or it is even essential for activity. Unfortunately, the analysis of membrane protein oligomerization *in vivo* is challenging due to the high complexity of natural membranes. Furthermore, classical *in vitro* studies need to implement disruption of membranes and subsequent solubilization of the membrane proteins into detergents (*Seddon et al., 2004*). This relatively harsh treatment can irreversibly modify the stoichiometry or even disrupt and unfold native membrane protein complexes. In addition, the presence and composition of the lipid environment can influence folding, proper adoption of quaternary structures and the assembly of heteromeric complexes. Lipids can even be components of membrane protein structures and act as modulators of membrane protein function (*van Dalen et al., 2002*; *Hunte and Richers, 2008*; *Laganowsky et al., 2014*). In this respect, the recently developed nanodisc technology, based on well-defined disc-shaped bilayer patches stabilized by two copies of a membrane scaffold protein, provides excellent and well-defined alternative lipid environments for membrane proteins (*Bayburt and Sligar, 2010*) (*Figure 1*). The utilization of nanodiscs is extremely beneficial as they are

**Figure 1.** Processes for membrane protein complex characterization. (**a** and **b**) Workflows implementing cellular expression and detergent solubilization of membrane proteins. (**a**) Membrane protein analysis with non-MS-related methods. (**b**) Membrane protein analysis with ESI-MS. (**c**) Proposed method combining cell-free co-translational membrane insertion with LILBID-MS analysis. Detergents and lipids are highlighted in orange and grey, respectively. The membrane scaffold protein is represented by blue helices and membrane proteins with red helices. The critical membrane extraction step is emphasized by the red box.

completely soluble, enabling the application of numerous biophysical and biochemical techniques for the characterization of membrane proteins in a native bilayer environment (*Denisov and Sligar, 2016*). Furthermore, in order to avoid potentially critical detergent contacts of membrane proteins, we have developed a detergent-free strategy for the co-translational insertion of membrane proteins into preformed empty nanodiscs by cell-free (CF) expression (*Roos et al., 2014*). For example, in the case of the extremely detergent-sensitive *E. coli* MraY translocase, a membrane-integrated enzyme essential for lipid I biosynthesis, this lipid-based CF expression (L-CF) was the only way to obtain functionally folded protein (*Ma et al., 2011*; *Henrich et al., 2016*). In addition, CF expression enables the production of toxic or otherwise difficult membrane proteins and provides numerous options for their efficient labeling (*Reckel et al., 2010*; *Henrich et al., 2015b*).

Most techniques employed to analyze protein complexes such as cross-linking, fluorescence spectroscopy or electron paramagnetic resonance spectroscopy frequently involve extensive protein modifications. However, oligomeric states of proteins can be extremely sensitive toward covalent modifications or environmental parameters (*Politi et al., 2009*; *Maciejko et al., 2015*). Recently developed non-covalent mass spectrometry (MS) using mild ionization techniques such as electrospray-ionization (ESI) or laser-induced-liquid-bead-ion-desorption (LILBID) thus became key technologies for the analysis of soluble proteins as well as of detergent solubilized membrane proteins (*Barrera and Robinson, 2011*). Sample transfer from solution to gas phase is most critical for noncovalent MS and requires carefully adjusted settings to maintain intact membrane protein complexes (*Barrera et al., 2008*). LILBID-MS as a technique complementary to ESI-MS emerged as new tool to investigate membrane proteins in native-like environments, especially based on its wide tolerance to additives needed for membrane protein solubilization (*Morgner et al., 2007*; *Vonck et al., 2009*; *Maciejko et al., 2015*).

The approach to employ ESI-MS in combination with nanodiscs has recently been successful to investigate membrane proteins in a native-like lipid environment (*Hopper et al., 2013*; *Marty et al., 2016*) (*Figure 1b*). However, this procedure still includes the risk of complex disintegration by the classical detergent treatment of the protein samples (*Figure 1b*). By combining cell-free synthetic biology (*Hodgman and Jewett, 2012*; *Smith et al., 2014*), nanodiscs and LILBID-MS, we have

established a new platform avoiding any detergent contacts and enabling the analysis of non-modified membrane proteins in their initial expression environment (*Figure 1c*). In addition, the created synergies streamline the process of membrane protein complex analysis by reducing the workflow to three basic steps (*Figure 1c*). The platform is suitable for a wide range of applications as it extends the analysis of challenging membrane proteins (i) that cannot be synthesized in vivo, (ii) that are sensitive to detergents, (iii) that do not tolerate covalent modifications or (iv) that require stable isotope labeling for better resolution.

We demonstrate the versatility of our approach by analyzing oligomeric states of six different membrane proteins from different functional classes and by gaining first insights indicating a lipid-dependent dimerization of *E. coli* MraY translocase.

## Results

### LILBID-MS analysis of empty nanodiscs

Initially, we investigated empty nanodiscs consisting only of the scaffold protein and lipids to define the individual particle behavior. When the LILBID laser, desorbing the sample droplets, was tuned to a low laser power of 10 mJ, signals were revealed corresponding to intact nanodiscs comprising two copies of the scaffold protein and roughly 200 DPPC lipids (*Figure 2a*). However, spectra remain broad and not well resolved and are not suitable for detailed assignments. Nevertheless, single lipids could be resolved by tuning the laser power to 23 mJ causing the disintegration of the nanodiscs. Detected signals correspond to scaffold proteins with a few remaining lipid molecules. A complete overview of all analyzed empty nanodiscs is shown in *Table 1*. The homogeneity of the analyzed nanodiscs was controlled by size exclusion chromatography (*Figure 2—figure supplements 1b*, *2b* and *3b* and *Figure 3b*) and exemplarily by negative stain electron microscopy (*Figure 2—figure supplement 1b*).

In contrast to previous ESI-MS-studies (*Marty et al., 2012*), intact nanodiscs were hardly detected at the refined LILBID settings necessary to achieve resolved signals (*Figure 2—figure supplement 1–3*). To investigate the influence of different nanodisc features on the obtained mass spectra, we scanned various scaffold proteins and lipids. Switching from MSP1D1ΔH5 over MSP1 to the larger MSP1E3D1 scaffold protein coincides with an increase from six to up to eleven attached DMPG molecules (*Figure 2—figure supplement 1a*, *Table 1*). Elongation of the alkyl chain with unsaturated alkyl groups led to further increased numbers of attached lipids with 15 or 20 for POPG and DOPG, respectively (*Figure 2—figure supplement 2a*, *Table 1*). In contrast to the chain length, the variation of the lipid head groups (from DOPG to DOPS or DOPC) did not modulate lipid attachments drastically (*Figure 2—figure supplement 3a*, *Table 1*). Notably, the utilization of DOPE lipids leads to the formation of highly heterogeneous nanodiscs (*Figure 2—figure supplement 3a,b*). This could be caused by the intrinsic tendency of PE lipids to form non-bilayer phases with higher curvature (*Gruner, 1985*), which possibly prevents the formation of proper nanodiscs. Moreover, as shown for the multi-drug transporter EmrE, non-bilayer lipids can significantly decrease reconstitution and activity of membrane proteins in liposomes (*Curnow et al., 2004*). Due to these possible negative effects on disc stability, nanodiscs reconstituted with DOPE were excluded from further studies.

More complex lipid compositions as present in native membranes were mimicked by mixtures including lipids such as cholesterol or cardiolipin (*Figure 3*). In case of DMPG:cardiolipin mixtures, the mass difference between the two lipids was sufficient to clearly distinguish and count the scaffold-bound cardiolipin and DMPG molecules (*Figure 3c*). It appears therefore feasible to tune the complexity of resulting mass spectra via specific nanodisc selection as well as to identify lipids staying attached to proteins within the sample. In general, spectra of nanodiscs composed out of lipids with shorter chain length lead to better resolved spectra due to the decreased number of attached lipid molecules. This will be especially important the more complicated the spectra of membrane protein/nanodiscs complexes are. Therefore, nanodiscs containing the short-chain lipids DMPG and DMPC were chosen for the following studies of membrane protein/nanodiscs complexes.

### Oligomeric state analysis of membrane proteins in nanodiscs

To establish the reliability of the method for analyzing membrane proteins from nanodiscs, we started with well-studied model systems with known subunit stoichiometry (*Table 2*). The potassium

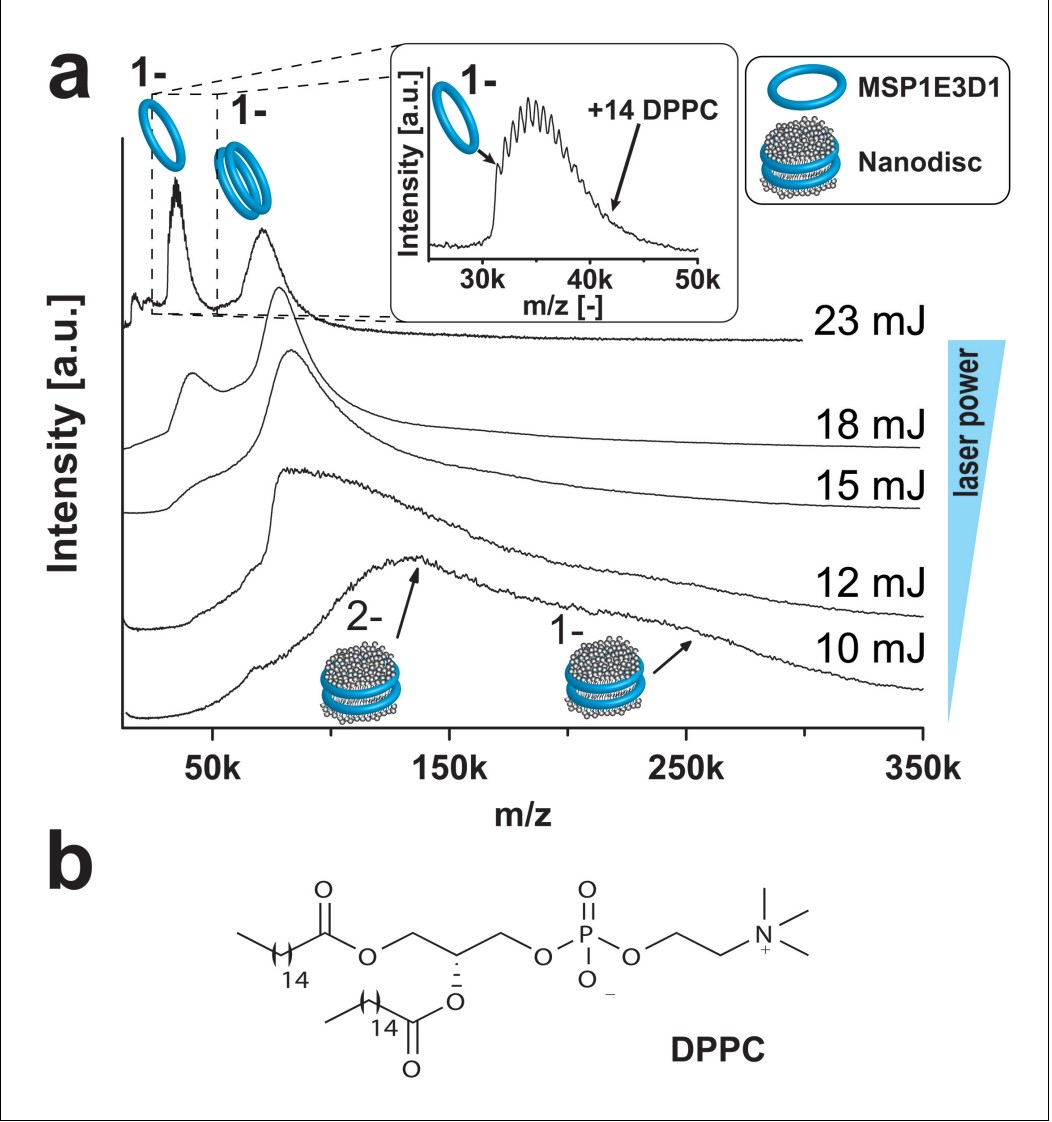

**Figure 2.** Adjusting LILBID-MS conditions for nanodiscs analysis. Pictograms illustrate complexes of the corresponding peaks and are labeled with the particle charge in the gas-phase. (a) The different laser intensities are indicated. The monomer region of MSP1E3D1 with lipid attachments is enlarged in the inset. (b) Structure of the lipid DPPC used for nanodisc formation.

The following figure supplements are available for figure 2:

**Figure supplement 1.** LILBID-MS analysis of nanodiscs assembled with different scaffold protein derivatives.

**Figure supplement 2.** LILBID-MS analysis of MSP1E3D1 nanodiscs assembled with lipids containing different chain length and saturation state.

**Figure supplement 3.** LILBID-MS analysis of MSP1E3D1 nanodiscs assembled with lipids containing different head groups.

channel KcsA (*Cortes and Perozo, 1997*; *van Dalen et al., 2002*), the small multidrug transporter EmrE (*Ubarretxena-Belandia et al., 2003*; *Morrison et al., 2012*) (*Figure 4a*) and the signal pepti-dase LspA (*Vogeley et al., 2016*) are known to be tetramer, dimer and monomer, respectively. The complexes were co-translationally inserted into nanodiscs assembled with the scaffold protein

**Table 1.** Characteristics of nanodiscs and contained bilayer for used combinations of different scaffold proteins with various lipids.

| Scaffold | Nanodisc diameter [nm] | Lipid | Protein:lipid ratio in nanodisc reaction mix | Attached lipids (LILBID) | Transition temparature [°C] | Headgroup charge $\Sigma$ charges | Hydrophobic tail | |
|---|---|---|---|---|---|---|---|---|
| | | | | | | | Double bonds | Length |
| MSP1D1ΔH5 | 8 | DMPC | 1:50 | 15 | 24 | 0 | 0 | 14 |
| | | DMPG | 1:45 | 6 | 23 | −1 | 0 | 14 |
| MSP1 | 10 | DMPC | 1:80 | 17 | 24 | 0 | 0 | 14 |
| | | DMPG | 1:70 | 9 | 23 | −1 | 0 | 14 |
| MSP1E3D1 | 12 | DMPC | 1:115 | 19 | 24 | 0 | 0 | 14 |
| | | DMPG | 1:110 | 11 | 23 | −1 | 0 | 14 |
| | | DPPC | 1:100 | 18 | 41 | 0 | 0 | 16 |
| | | POPG | 1:90 | 15 | −2 | −1 | 0/1 | 16/18 |
| | | DOPC | 1:80 | 21 | −17 | 0 | 2 | 18 |
| | | DOPG | 1:80 | 20 | −18 | −1 | 2 | 18 |
| | | DOPS | 1:90 | 20 | −11 | −1 | 2 | 18 |
| | | DOPE | 1:80 | 8 | −16 | 0 | 2 | 18 |
| | | Mixtures | | | Lipid ratio | | | |
| | | DMPG/ DMPC | 1:115 | 11 | 50/50 | −1/0 | 0/0 | 14/14 |
| | | DMPG/ Cardiolipin | 1:120 | 12 | 90/10 | −1/−2 | 0/0 | 14/14 |
| | | DMPG/ Inositol | 1:110 | 15 | 50/50 | −1/−1 | 0/0;1 | 14/ 16;18 |
| | | DMPC/ Cholesterol | 1:115 | 23 | 90/10 | 0/0 | 0/ n.d. | 14/ n.d. |
| | | E. coli total lipids | 1:50 | n.i. | n.d. | n.d. | n.d. | n.d. |

n.d. = not defined.

n.i. = not investigated.

MSP1E3D1 and either DMPG or DMPC lipids. The big nanodiscs containing MSP1E3D1 were chosen as they performed best in previous studies regarding membrane protein solubilization (*Roos et al., 2012*).

For all proteins, clear MS signals were detected corresponding to the expected oligomeric states in combination with the two membrane scaffold proteins (MSPs) (*Figures 4b,c* and *5a*). Elevated laser energies revealed additional signals assigned to the respective complexes after loss of MSPs or target complex subunits. *Figure 4b* clearly shows signals consistent with EmrE monomer and dimer without any attached scaffold proteins, indicating that within LILBID-MS analysis the EmrE complex can be dissociated from the nanodiscs without losing its integrity. Similar signals for KcsA tri- and tetramer without scaffold proteins cannot yet be unambiguously assigned due to higher complexity of the sample causing overlap with other species (*Figure 4c* (ii)). Further modifications that allow successful assignment will be explained in the next section. However, the tetrameric stoichiometry deduced from the low-laser intensity spectra could be confirmed by SDS-PAGE (*Cortes and Perozo, 1997*) (*Figure 4d*). Comparison of EmrE spectra taken in LILBID anionic as well as in cationic mode revealed similar complexes (*Figure 4—figure supplement 1*). As spectra appeared to be better resolved in the anionic mode, this type was chosen for all further measurements.

Besides these two well-characterized model systems, we further analyzed nanodiscs containing the voltage-sensing domain of the human proton channel Hv1 (hHv1-VSD). In this case, the nanodiscs were assembled using the smaller MSP version MSP1D1ΔH5 as the preparation of cell-free produced VSD samples has already been optimized for NMR investigations (*Laguerre et al., 2016*). We could reveal a dimeric state (*Figure 5b*), despite the lack of the intracellular domain being

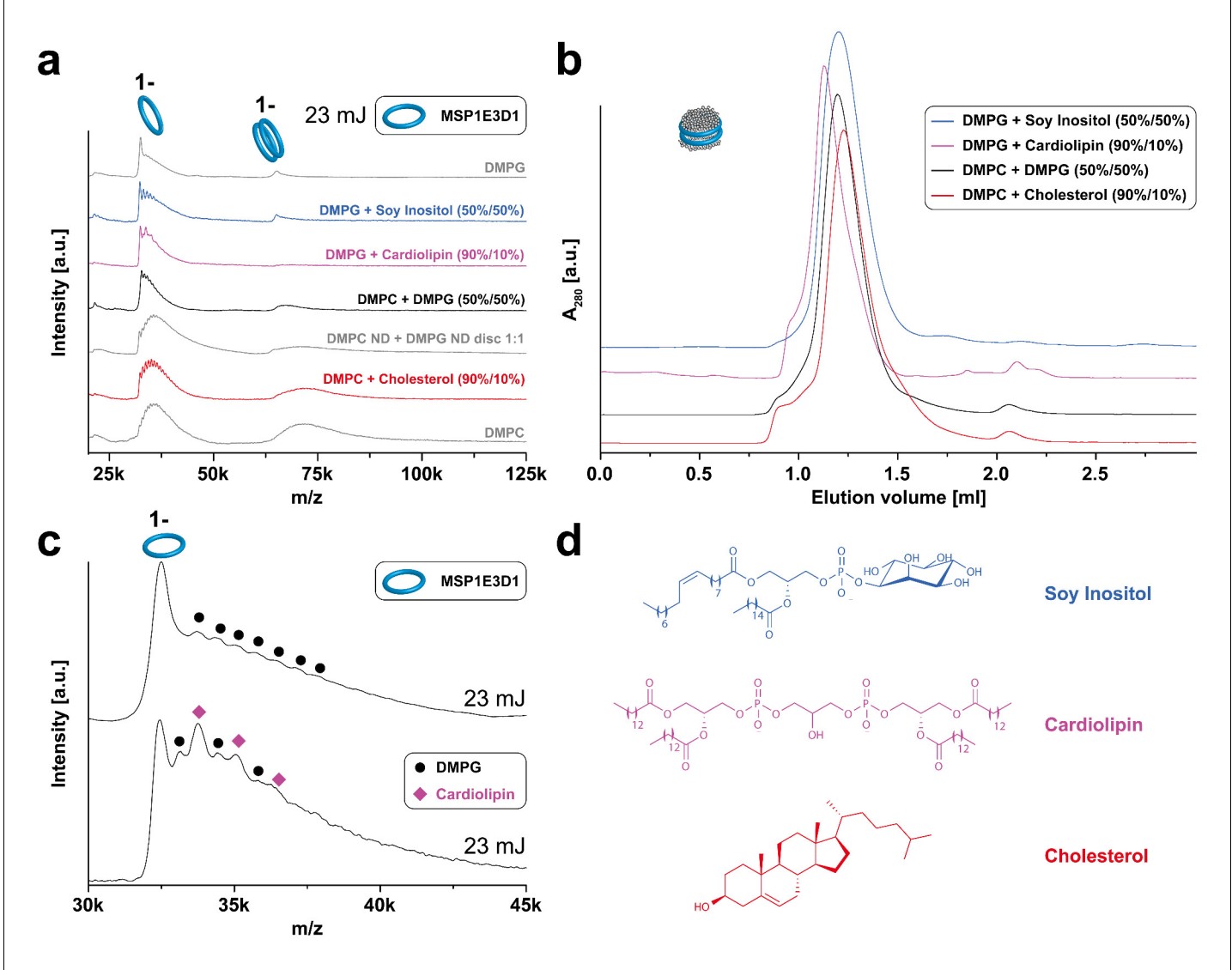

**Figure 3.** LILBID-MS analysis of MSP1E3D1 nanodiscs assembled with lipid mixtures. Pictograms illustrate the detected particles. The applied laser intensity is indicated. (a) LILBID-MS spectra of nanodiscs containing the indicated lipid mixtures as well as single lipid types. (b) Corresponding size exclusion profiles of the nanodiscs on a Superdex 200 3.2/30 increase column with a flow rate of 0.075 µl/min. (c) LILBID-MS spectra of MSP1E3D1 (DMPG) (top) and MSP1E3D1 (DMPG/cardiolipin) nanodiscs (bottom). Circles and squares highlight peaks generated by DMPG or cardiolipin molecules attached to MSP1E3D1 monomers. (d) Structures of lipids used for nanodisc formation.

proposed as the main dimerization interface. Thus, our strategy supports recent findings that this domain is not essential for hHv1 dimerization (*Li et al., 2015*).

## Refinement of LILBID-MS signal resolution by isotope labeling

As expected, signal overlap of different fragments upon increased sample complexity can limit the evaluation and assignment of our spectra. In *Figure 4c*, the signals of the free KcsA tri- and tetramer fragments without attached scaffold proteins are ambiguous as they are too close in mass to other potential species. As an example, the 85.32 kDa KcsA tetramer differs only by 0.07 kDa from a complex of two MSP1E3D1 scaffold proteins with one KcsA monomer. An efficient approach for minimizing such overlap is labeling of the membrane proteins with stable isotopes such as [15]N, [13]C and/or [2]H, which is a key advantage of the implemented cell-free expression technique (*Sobhanifar et al., 2010*).

**Table 2.** Construct list combining full name, size and expression system with the abbreviation. The theoretical masses of the different constructs are calculated by the Expasy tool protparam according to the primary sequence. Based on that the theoretical molecular weights of labeled proteins are calculated by the addition of masses of the single amino acids subtracted by the number of water molecules corresponding to the number of peptide bonds.

| Protein | Molecular weight [kDa] | | Full name | Reference |
| | Theoretical | Experimental | | |
| --- | --- | --- | --- | --- |
| PR | 27.09* | 27.19* | Proteorhodopsin (green variant) | (*Reckel et al., 2011*) |
| KcsA | 21.33/22.70[†] | 21.38/22.77[†] | pH-gated potassium channel KcsA | - |
| Emre | 13.00 | 13.09 | Multidrug transporter EmrE | (*Roos et al., 2012*) |
| LspA | 21.52/22.83[†] | 21.63/22.84[†] | Lipoprotein signal peptidase | (*Laguerre et al., 2016*) |
| hHv1-VSD | 20.80/22.28[†] | 20.89/22.20[†] | Human proton channel - voltage sensing domain | (*Li et al., 2015*) |
| Bs-MraY | 37.88 | 37.93 | Phospho-N-acetylmuramyl-pentapeptide-transferase *B.subtilis* | (*Henrich et al., 2016*) |
| Ec-MraY | 42.22 | 42.29 | Phospho-N-acetylmuramyl-pentapeptide-transferase *E.coli* | (*Henrich et al., 2016*) |
| MSP1 | 25.3 | 25.40 | Membrane scaffold protein | (*Denisov et al., 2004*) |
| MSP1E3D1 | 31.96 | 32.08 | Membrane scaffold protein | (*Denisov et al., 2004*) |
| MSP1D1ΔH5 | 19.49[‡]/21.46 | 19.53[‡]/21.50 | Membrane scaffold protein | (*Hagn et al., 2013*) |

\* Molecular mass according to covalent coupling of all trans-retinal.

† Molecular mass according to heavy isotope labeling.

‡ Molecular mass according to cleavage of the His$_6$-tag.

As expected, the $^{15}$N and $^2$H isotope labeling of KcsA in MSP1E3D1 nanodiscs improved spectral resolution and revealed signals for the free tetramer at 91.08 kDa, now separated by 4.14 kDa from the KcsA monomer with two attached MSPs detected at 86.93 kDa (*Figure 5c*).

Similarly, ambiguity is also present in the spectra of hHv-1-VSD (*Figure 5b*). The scaffold protein MSP1D1ΔH5 without the implemented His$_6$-tag has a molecular mass of 19.49 kDa and is therefore close to the hHv1-VSD monomer (20.80 kDa). Again, $^{15}$N and $^2$H labeling of hHv1-VSD separates both signals clearly by increasing the mass of hHv1-VSD to 22.20 kDa, now resulting into a mass difference of 2.71 kDa. For KcsA and hHv1-VSD, the separation of signals by heavy isotope labeling was sufficient to reveal the different oligomeric states despite the attachments of lipids. Unfortunately, similar lipid attachments prevented the clear separation of the signals of uncleaved MSP1D1ΔH5 at 21.46 kDa from that of a signal peptidase monomer LspA at 22.84 kDa by the $^{15}$N, $^2$H labeling approach (*Figure 5a*). However, signal overlap can alternatively be addressed by using different scaffold protein derivatives for nanodiscs assembly. Translation of unlabeled LspA (21.52 kDa) into MSP1E3D1 nanodiscs instead of MSP1D1ΔH5 nanodiscs sufficiently separated the signals now revealing the expected monomeric state of LspA.

## Oligomer formation of proteorhodopsin (PR) in nanodiscs

In order to explore the limits of our strategy, we analyzed the seven transmembrane helix green-light absorbing PR, a light-driven retinal containing proton pump. PR forms up to hexameric complexes in its natural membrane environment but is also stable and active as a monomer (*Klyszejko et al., 2008*; *Reckel et al., 2011*; *Maciejko et al., 2015*). Single-molecule force microscopy with cell-free synthesized PR/nanodisc samples already suggested multimeric conformations of PR co-translationally inserted into MSP1E3D1 nanodiscs (*Roos et al., 2012*).

Monomeric PR occupies a surface of around ~6 nm$^2$. A pentameric or hexameric complex of PR would thus potentially require a membrane space of at least 25–30 nm$^2$ and 33–36 nm$^2$, respectively (*Klyszejko et al., 2008*). It should therefore be possible that even such rather large complexes can assemble within nanodiscs composed of MSP1E3D1 having a calculated membrane area of 113 nm$^2$ (*Denisov et al., 2004*). As proposed and in agreement with previous findings obtained with complementary techniques, the MS analysis under mild laser conditions of cell-free generated PR/nanodiscs complexes composed of MSP1E3D1 and DMPG revealed multimeric states of PR. The spectra show a distribution of complexes containing two scaffold proteins and up to six PR. Elevated laser

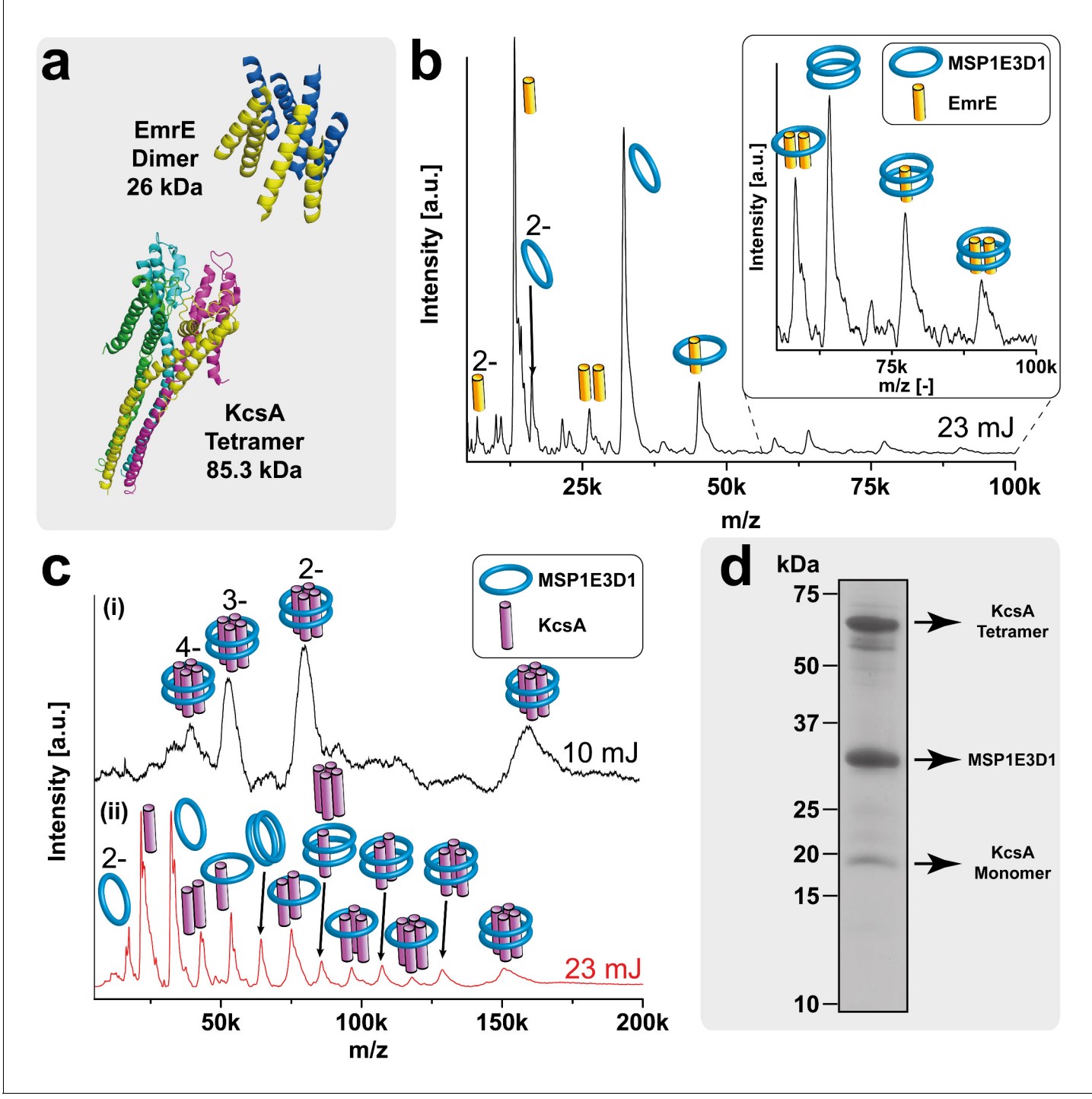

**Figure 4.** LILBID-MS analysis of KcsA and EmrE nanodisc complexes. Pictograms in mass spectra illustrate the assigned complex for every peaks. Charge states are indicated if they differ from 1-. The different laser intensities are indicated. (**a**) 3D structures and mass of KcsA (PDB: 3EFF) and EmrE (PDB: 2I68) (**b**) Mass spectrum of EmrE MSP1E3D1 (DMPG) complexes. Free monomer and dimer can be dissociated from the scaffold proteins. (**c**) KcsA MSP1E3D1 (DMPG) complexes analyzed at different laser conditions. Under soft conditions, (i) the KcsA tetramer with two scaffold proteins appears with one to four charges. Higher desorption laser energy (ii) leads to dissociation of scaffold or KcsA proteins. Monomer and dimer of KcsA without scaffold proteins are visible, but free tri- and tetramer cannot yet be unambiguously assigned due to overlap with other species. (**d**) SDS-PAGE analysis of the KcsA MSP1E3D1 (DMPG) sample allowed verifying the formation of SDS stable tetramer.

The following figure supplement is available for figure 4:

*Figure 4 continued on next page*

*Figure 4 continued*

**Figure supplement 1.** LILBID-MS analysis of nanodiscs with different ionization modes.

conditions revealed various additional signals of PR with or without scaffold proteins indicating the formation of stable PR complexes up to penta- and hexamers (*Figure 6a*). In accordance to the results obtained with KcsA and EmrE, this data further confirms the capability of the implemented cell-free expression approach to generate native higher oligomers of diverse membrane proteins by co-translational insertion into nanodiscs.

But what is the significance of the different observed species? To interpret these it is helpful to compare KcsA and PR. Mild laser conditions revealed only tetrameric complexes for KcsA (*Figure 4c*), but a variety of oligomeric states for other proteins such as PR (*Figure 6a*). A possible explanation could be laser-induced dissociation of protein complexes, with the degree of dissociation depending on the particular complex stability. Detection of several oligomeric forms could also reflect a natural variation in complex formation. PR is folded and active under numerous conditions such as pentamers or hexamers, and its characteristic photocycle is observed even in monomeric conformation (*Reckel et al., 2011*). However, we have seen that, if laser dissociation occurs, the scaffold proteins are among the first proteins to leave the disc (*Figure 6*), giving rise to signals representing complexes with only one or no scaffold proteins. As the dominating species of the heterogeneous distribution of PR oligomerization states under mild laser conditions all include both scaffold proteins we can conclude that the detected species represent complexes in solution, meaning that PR is present in different oligomeric states. Under high laser intensities PR oligomers without scaffold proteins are released and can be detected as well up to hexamers, indicating that the PR proteins indeed form complexes and are not present as individual entities in the nanodiscs.

The potential importance of the lipid environment for protein oligomerization is further demonstrated by the comparison of cell-free expressed PR either in the presence of nanodiscs or detergents. Protein synthesized in the presence of the detergents DH7PC and GDN, previously identified to support the folding and stability of PR, was mostly present in monomeric conformation (*Figure 6—figure supplement 1a*). This observation is in agreement with previous structural studies with solution NMR by using identical sample preparations and showing monomeric PR (*Reckel et al., 2011*). In contrast to the utilization of different detergents that results in different oligomer distributions (*Reckel et al., 2011*; *Maciejko et al., 2015*), the variation of the lipid within the nanodiscs bilayer does not change the oligomerization behavior of cell-free produced PR (*Figure 6—figure supplement 1b*). In the case of PR lipids seem to support the formation of higher oligomer complexes but PR does not discriminate between different lipid species. Due to the increased number of attached lipids for other lipid species as discussed before for empty nanodiscs the peaks get broader and less resolved, but the overall peak pattern stays the same indicating a comparable oligomeric composition (*Figure 6—figure supplement 1b*).

## Structure-function relationship of MraY

The membrane-integrated enzyme MraY is essential for the bacterial peptidoglycan biosynthesis by linking cytoplasmic precursors to membrane-bound lipid carrier molecules resulting in lipid I formation. Like LspA, MraY is a potential target for drug design to tackle multiple-resistant bacteria (*Bugg et al., 2016*). In contrast to PR which's oligomer formation is not discriminated by the bilayer environment, the activity of MraY homologues is remarkably dependent on specific lipid requirements (*Roos et al., 2012*; *Henrich et al., 2016*). While *B. subtilis* MraY activity is not lipid selective, the activity of MraY homologues from Gram-negative bacteria such as *E. coli* requires anionic lipids like DMPG (*Henrich et al., 2016*). In respect to the formation of oligomers of MraY two-hybrid studies (*White et al., 2010*), crosslinking experiments and the crystal structures of MraY (*Chung et al., 2013*, *2016*) indicate a dimeric state, but correlations of the oligomeric state and catalytic activity within different lipid environments have not yet been studied.

We have analyzed the oligomeric states of *B. subtilis* MraY and of *E. coli* MraY in MSP1E3D1 nanodiscs assembled with either anionic or non-charged zwitterionic lipids. *B. subtilis* MraY always revealed a dimeric state within both lipid environments (*Figure 7a*). In contrast, *E. coli* MraY shows

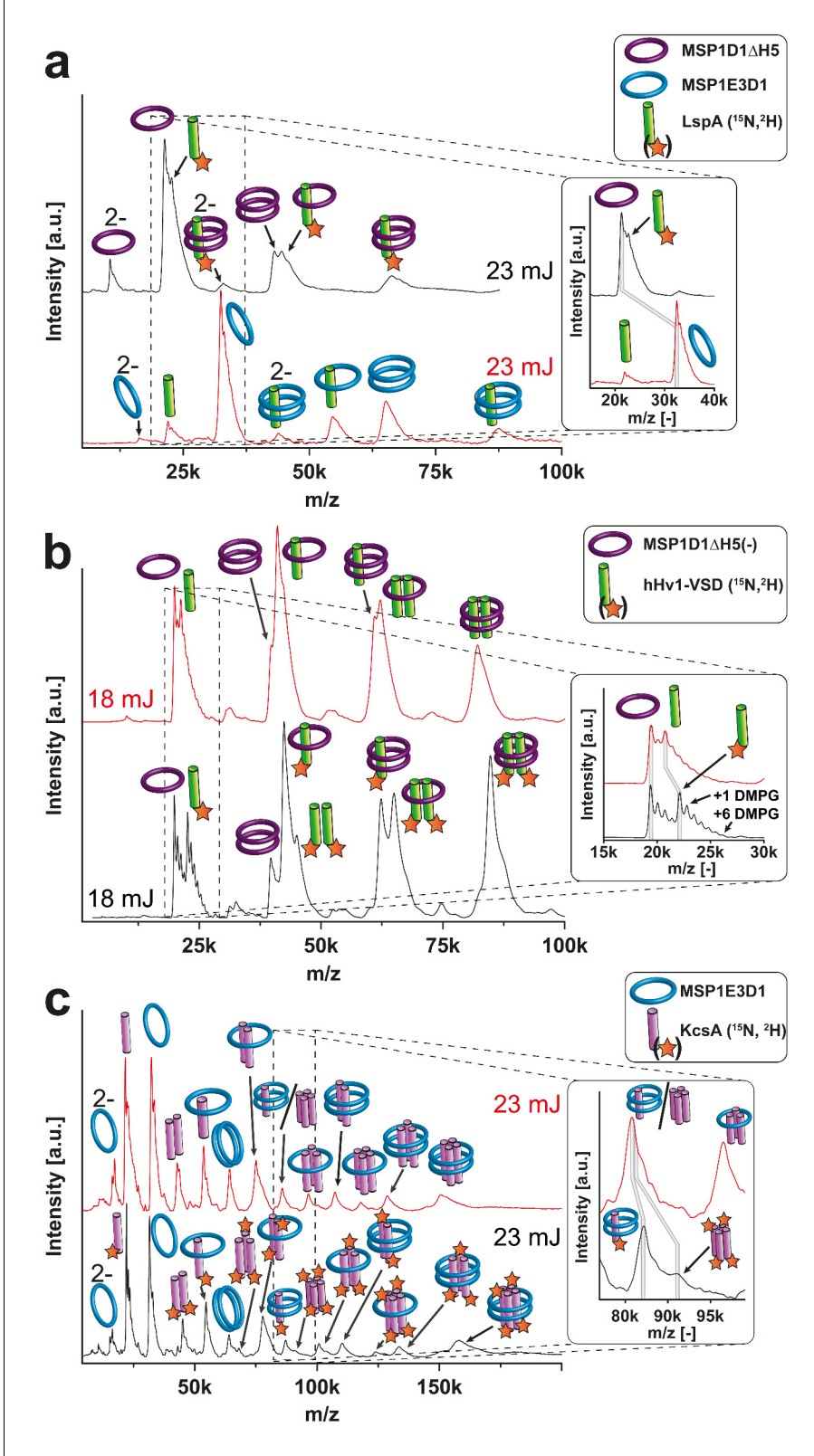

**Figure 5.** Resolution refinement by isotopic labeling or scaffold protein exchange. Relevant mass shifts reducing the peak overlap are highlighted in the insets. Pictograms illustrate the corresponding complex of detected peaks and asterix highlight heavy isotope-labeled protein as indicated. The different laser intensities are indicated. (a and c) Spectra of labeled and non-labeled samples are shown by a black and a red line, respectively. (a) hHv1-VSD in nanodiscs using MSP1D1ΔH5 scaffold proteins (19.49 kDa). Labeling shifts the hHv1-VSD monomer about 2 kDa. (b) KcsA MSP1E3D1

*Figure 5 continued on next page*

*Figure 5 continued*

(DMPG) complexes. Mass shift for the free KcsA tri- and tetramer (3.63 and 4.48 kDa, respectively) are sufficient to stop overlap with other species. (c) Spectra of LspA MSP1D1ΔH5 (DMPC) complexes in black and of LspA MSP1E3D1 (DMPG) complexes in red. $^{15}$N, $^{2}$H labeled LspA (22.57 kDa) could not be sufficiently separated from MSP1D1ΔH5 (21.46 kDa). However, spectra of unlabeled LspA (21.52 kDa) inserted into MSP1E3D1 discs show no overlap.

dimer formation only if inserted into membranes composed out of the anionic lipid DMPG, but it remains monomeric in DMPC (*Figure 7b*). Interestingly, a signal for the dimeric state of *E.coli* MraY could be restored by using nanodiscs containing an equimolar mixture of DMPC and DMPG (*Figure 7b*). The observed dimer formation exactly correlates with the previously reported enzymatic activity in the different membranes (*Figure 7c*) (*Henrich et al., 2016*), suggesting for the first time that the active conformation of MraY could be a dimer and that this conformation is triggered by the composition of the bilayer. A lipid-like electron density in the crystal structure of *Aquifex aeolicus* MraY located in a hydrophobic tunnel within the dimer interface (*Chung et al., 2013*) (*Figure 7d*) further supports a potential role of lipids for MraYs structural and functional integrity.

## Discussion

While detergents are well-established environments for the analysis of membrane proteins and their non-covalent interactions, they can drastically affect conformation and stoichiometry of membrane protein complexes. The presented strategy is therefore complementary to established techniques as it allows a completely detergent-free process suitable also for detergent-sensitive membrane

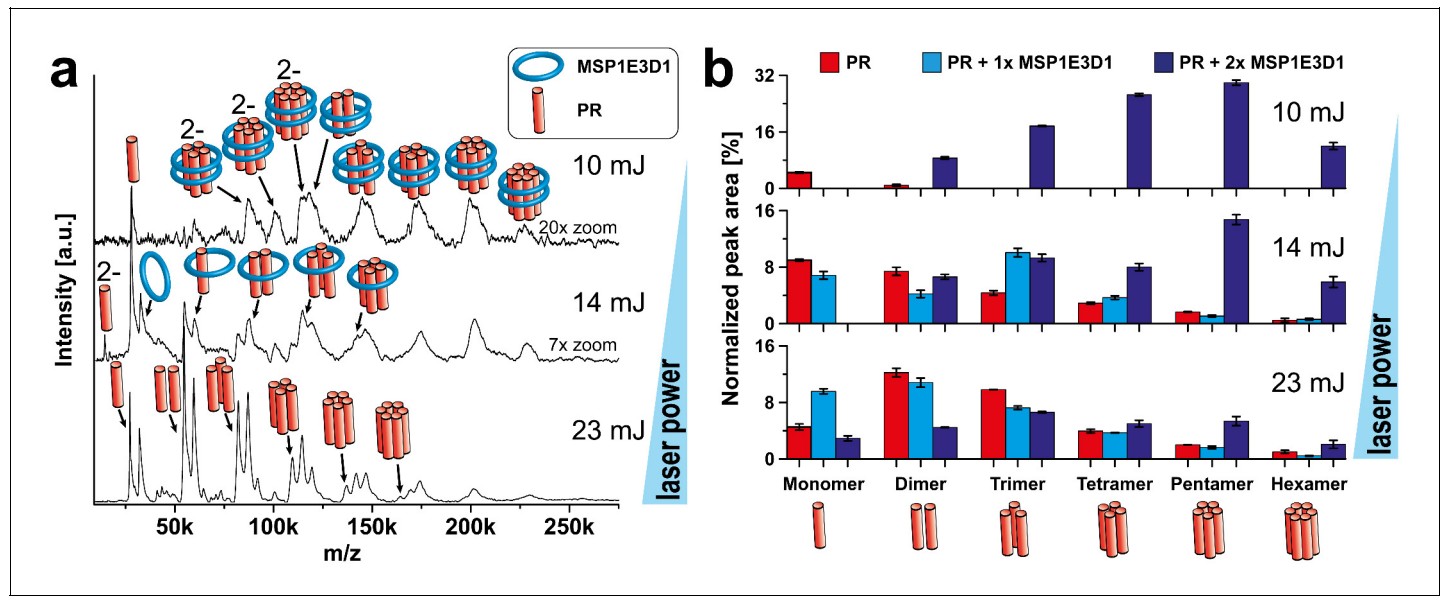

**Figure 6.** Complex analysis of PR with varying laser conditions. LILBID-MS spectra of cell-free expressed PR with increasing laser power (a) and corresponding statistical analysis of the abundance of different complex species (b). The different laser intensities are indicated and pictograms illustrate the corresponding complex of the detected peaks. (a) PR in nanodiscs composed of MSP1E3D1 and DMPG shows up to hexameric complexes with two attached scaffold molecules. With elevated laser powers, these complexes get more and more disrupted and plain PR complexes without scaffold attachments can be detected. This is nicely illustrated by the statistical analysis (b) of the spectra shown in (a). Using low-laser intensities mainly PR complexes associated to scaffold proteins can be detected. The values of the bars correspond to the means of 2–3 measurements and the error bars display the standard deviation.

The following figure supplement is available for figure 6:

**Figure supplement 1.** Analysis of PR expressed into different hydrophobic environments.

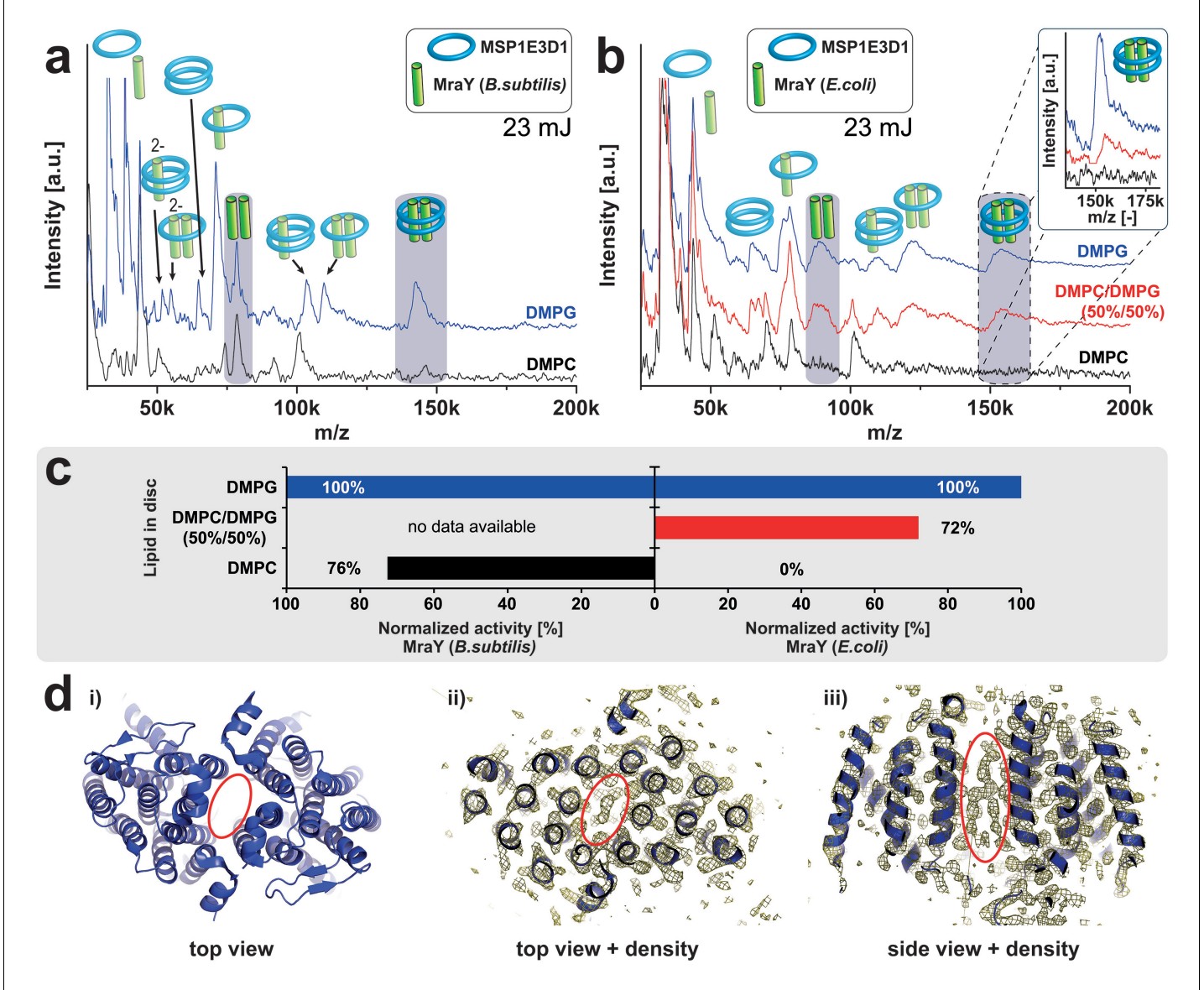

**Figure 7.** Lipid related effect on structure and function of *E. coli* MraY. LILBID-MS spectra of *B. subtilis* MraY (**a**) and *E. coli* MraY (**b**) in MSP1E3D1 nanodiscs with different lipids. Pictograms illustrate the composition of detected signals and lipids for each sample are indicated. Grey boxes in spectra highlight MraY dimer signals with and without scaffold protein. The different laser intensities are indicated. The inset shows a close-up of the MraY dimer region with scaffold measured under high-mass conditions. (**c**) Enzymatic activity of *B. subtilis* and *E. coli* MraY in nanodiscs with corresponding lipids (*Henrich et al., 2016*). (**d**) 3D crystal structure of *A. aeolicus* MraY (PDB: 4 J72). Electron density maps of top and side views indicate a hydrophobic tunnel within the dimer interface potentially filled with lipid molecules as highlighted by a red ellipse (*Chung et al., 2013*).

proteins. Moreover, our platform provides valuable synergies by combining (i) fast and efficient production of even toxic or specifically labeled membrane proteins enabled by cell-free expression, (ii) defined, stable membrane environments contributed by nanodiscs and (iii) sensitive and tunable particle detection facilitated by LILBID-MS. This strategy is, furthermore, a fast and straightforward process and does not depend on chemical protein modifications.

The complexity of MS spectra of membrane protein/nanodiscs complexes is increased by the presence of the scaffold protein that can impede the clear identification of membrane protein signals. Labeling of the target protein with stable isotopes can address this problem as demonstrated for KcsA or hHv1-VSD. The sample production by cell-free expression is here highly advantageous,

as the complete control over low-molecular-weight compounds in cell-free systems allows efficient labeling of synthesized proteins in any amino acid combination (*Reckel et al., 2010*; *Sobhanifar et al., 2010*; *LaGuerre et al., 2015*). Switching to a different scaffold protein upon nano-discs preparation could be an alternative option to improve spectral resolution as demonstrated for LspA. In general, even complicated LILBID-MS spectra are comparably straight forward to analyze. Despite the complexity caused by numerous oligomeric states of the target protein in addition to its combinations with different numbers of the scaffold protein, within LILBID-spectra every species is only present with a few charge states (*Morgner et al., 2006*). In our nanodiscs spectra mainly the singly charged species are observed.

An important tool to probe complex stability as well as interactions between membrane proteins and/or lipids is the laser intensity of LILBID-MS. Elevated laser intensity modulates the LILBID process resulting into partly disintegrated nanodisc structures. This is presumably caused during the explosive expansion of the sample droplet after laser irradiation. Lipids are lost most likely due to deformation and subsequent rupture of the flat disc-shaped membrane. However, the mild ionization conditions of LILBID-MS are capable of preserving at least partly the initial membrane protein complex. The absence of scaffold proteins and lipids, furthermore, improves spectral resolution and helps to identify the intact as well as partly dissociated protein complexes. This is especially effective if the scaffold size is largely compared to the protein complexes (*Figure 5a*).

The presented platform appears to be well suited to address challenging biological questions. Complexes indicating an artificial stoichiometry different from the expected oligomeric state could never be found for any tested protein. The highest detected oligomeric state always corresponded to the previously determined native state of the target. The results, furthermore, indicate that the implemented L-CF expression of membrane proteins results into the assembly of native complexes. Therefore, the strategy allows to rapidly determine the oligomeric state of a protein within the nanodiscs in context of a membrane with defined lipid composition, giving important insights for the understanding of functional dynamics. Lipid-dependent oligomerization is a so far rarely observed regulatory mechanism of membrane proteins. The reported study on the *E.coli* MraY translocase is one of the first examples showing such a lipid-triggered dimerization. Notably, even close homologues such as *B. subtilis* can significantly differ in their oligomeric assembly mechanism and the LILBID-MS observations nicely match with previous reports on MraYs functional activity (*Henrich et al., 2016*).

Instrumental improvements will continuously contribute to higher resolution and will open the method to extended applications. One further direction could be the identification of specific and tightly bound lipids, which could affect structural and functional features of a membrane protein. According to the identification of lipid-protein interactions in mixed micelles (*Gault et al., 2016*) or tightly bound lipids by a delipidation strategy (*Bechara et al., 2015*), individual lipid types selectively interacting with a protein might be identified out of a provided mixture in nanodiscs membranes.

## Materials and methods

### Detergents and lipids

Polyethylene glycol P-1,1,3,3-tetramethyl-butylphenyl-ether (Triton X-100), sodium cholate, 1,2 diheptanoyl-*sn*-glycero-3-phosphocholine (DHPC), dodecyl phosphocholine (DPC), glyco-diosgenin (GDN), 1,2-Dimyristoyl-sn-glycero-3-phosphocholine (DMPC), 1-palmitoyl-2-oleoyl-sn-glycero-3-phospho-(1'-racglycerol) (POPG), 1,2-dioleoyl-sn-glycero-3-phosphocholine (DOPC), 1,2-dimyristoyl-sn-glycero-3-phospho-(1'-racglycerol) (DMPG), 1,2-dioleoyl-sn-glycero-3-phospho-(1'-rac-glycerol) (DOPG), 1,2-dioleoyl-sn-glycero-3-phosphoethanolamine (DOPE), 1,2-dioleoyl-*sn*-glycero-3-phospho-L-serine (DOPS), 1,2-dipalmitoyl-*sn*-glycero-3-phosphocholine (DPPC), 1',3'-bis[1,2-dimyristoyl-*sn*-glycero-3-phospho]-*sn*-glycerol (Cardiolipin), cholesterol.

Detergent stock solutions of DHPC, DPC and GDN were prepared with a concentration of 10% (w/v) and water as solvent. Lipid stock solutions were prepared with 50 mM of lipid and 100–300 mM of sodium cholate dissolved in water supported by vortexing and sonication in an ultrasonic bath at 37°C.

## DNA techniques

Introduction of a C-terminal StrepII-tag in the hHv1-VSD (75-223) construct in pET15b, described elsewhere (*Li et al., 2015*), was achieved by using QuickChange mutagenesis.

## Nanodisc preparation

The different membrane scaffold proteins were expressed and purified as mentioned elsewhere (*Denisov et al., 2004*; *Hagn et al., 2013*; *Henrich et al., 2015a*). Nanodiscs were formed by the dialysis of a mixture of MSPs and lipids in a defined protein to lipid ratio as described previously (*Roos et al., 2012*; *Henrich et al., 2016*). These ratios are critical for the reconstitution of homogenous nanodiscs. Furthermore, the transition temperature of the applied lipid species (*Table 1*) needs to be considered as disc formation below these temperatures can be affected. In brief, purified MSP and the desired lipids were mixed at the defined ratios, combined with 0.1% DPC and incubated for 1 hr at room temperature. For the nanodiscs formation, these mixtures were applied to dialysis bags with a MWCO of 12–14 kDa or Slide-A-Lyzer devices with a MWCO of 10 kDa and dialyzed against 5 l DF-buffer (40 mM Tris-HCl pH 8.0, 100 mM NaCl) for ~16 hr at room temperature. The buffer was exchanged two times, and dialysis was carried out for additional 16 hr each. Before concentration, nanodiscs solutions were centrifuged (22,000 xg, 20 min) to remove aggregates. Nanodiscs were concentrated to concentrations of 0.5–1.0 mM by multiple centrifugation steps (2000 xg, 15 min) using Centriprep concentrator devices (10 kDa MWCO). Concentrated nanodiscs were flash frozen using liquid nitrogen and stored at −80° C until usage. Buffer exchange of nanodiscs to MS-buffer (50 mM ammonium acetate pH 6.8) was achieved by dialysis against 5 l MS-buffer. The final concentration of empty discs samples was 30–50 μM.

## Cell-free protein production

The membrane protein production is based on a continuous exchange cell-free system (CECF) utilizing lysates from *E.coli* A19 cells and on T7-RNA polymerase transcription. T7-polymerase expression and purification, lysate preparation and CECF expression were performed as described previously (*Schwarz et al., 2007*; *Henrich et al., 2015a*). Heavy isotope labeling was achieved by exchanging solutions of unlabeled amino acids to amino acids including $^{15}$N and $^{2}$H nuclei. This solution is made of algal amino acid mixture $^{15}$N, d ($^{15}$N, $^{2}$H uniformly labeled) with a final concentration of 1.5 mg/ml containing 16 amino acids. For these 16 amino acids only non-exchanging protons are substituted to $^{2}$H. The residual four amino acids namely asparagine ($^{15}$N labeled), cysteine (unlabeled), glutamine ($^{15}$N labeled) and tryptophan ($^{15}$N labeled), which are not included in the algal mixture, were added to this mix to a final concentration of 1.5 mg/ml. 55 μl of reaction mixture was applied to Mini-CECF containers that were placed in the cavities of a 24-well cell-culture plate holding 825 μl of feeding mixture. Reaction and feeding mix were separated by a semipermeable membrane with a cut-off of 12–14 kDa. Plates were incubated at 30°C for 14–20 hr on a shaker at 200 rpm. For membrane protein solubilization, 10–100 μM of nanodiscs of different sizes assembled with the scaffolds MSP1D1ΔH5 or MSP1E3D1 were added to the reaction mixture. Nanodisc concentrations in the CF reaction mixtures were 20 μM (EmrE, LspA, KcsA), 60 μM (hHv1-VSD) and 80–100 μM for MraYs. For PR expressions, 10 μM of MSP1E3D1 nanodiscs were supplemented. The big scaffold protein MSP1E3D1 was chosen for most samples as it offers the most membrane space and was previously shown to exhibit the best protein solubilization (*Roos et al., 2012*). Exceptions were made in the case of LspA and hHv1-VSD as for these proteins the solubilization using nanodiscs containing MSP1D1ΔH5 as scaffold was already shown to be successful (*Laguerre et al., 2016*). Concentrations were chosen to achieve a high but not totally complete degree of membrane protein solubilization (*Roos et al., 2012*; *Henrich et al., 2016*). The reaction mixtures of PR expressions were supplemented with 600 μM all trans-retinal dissolved in DMSO. For detergent mode (D-CF) production of PR (*Reckel et al., 2011*), the reaction mix and the feeding mix contained 0.4% GDN and 0.04% DHPC. Volumes of reaction mixtures of samples dedicated for MS measurements ranged from 400 μl up to 1 ml.

## Purification of cell-free expressed membrane proteins

Unless stated otherwise, protein samples dedicated for mass spectrometry analysis were purified via the C-terminal StrepII-tag of the different membrane proteins. After expression, the reaction mix was centrifuged (22,000 xg for 10 min at 4° C) to remove residual aggregates and the supernatant

was diluted with buffer A (100 mM Tris-HCl pH 8.0, 100 mM NaCl) in the ratio 1:3. Subsequently, the solution was applied on equilibrated Streptactin resin with a bed volume corresponding to the reaction mix volume. The flow trough was re-applied three to five times to ensure maximum binding. After washing the material with six column volumes of buffer A, the proteins were eluted with 2–4 column volumes of buffer B (100 mM Tris-HCl pH 8.0, 100 mM NaCl, 20 mM desthiobiotin). For mass spectrometry analysis, the elution fractions were concentrated using Amicon Ultra centrifugal filters with 10 kDa MWCO to a final volume of 40–80 µl. Purification buffers for KcsA and MraY samples contained 20 mM KCl and 2 mM $\beta$-mercapotethanol, respectively. D-CF expressed PR was purified using the described protocol with the addition of 0.1% DHPC to all used buffers.

hHv1-VSD and LspA were purified with a tandem purification (*Laguerre et al., 2016*). $Ni^{2+}$-loaded sepharose resin was equilibrated with buffer C (100 mM Tris-HCl pH 8.0, 150 mM NaCl, 10 mM imidazole) and incubated in batch mode for 30 min at 4°C with the supernatant from the reaction mixture. After two washing steps with each 10 column volumes buffer C, followed by buffer C with 30 mM imidazole, the hHv1-VSD as well as LspA in nanodiscs were eluted with 250 mM imidazole in buffer C. The elution fraction was directly loaded onto the Strep-tactin resin and purified using the manual instructions from iba. All column steps were performed in gravity flow mode for hHv1-VSD and with an Äkta-Purifier system for LspA. The combined elution fractions were dialyzed against 20 mM Hepes pH 7.0, 20 mM NaCl over night with three buffer exchanges in case of hHv1-VSD. Buffer exchange for LspA into the MS-buffer was achieved in centrifugal concentration devices (10 kDa MWCO, Amicon). Protein solutions were concentrated as mentioned above.

The protein concentration was determined by $A_{280}$ measurements using a Nanodrop (*Roos et al., 2012*). As the sample consist of the target protein as well as of scaffold protein and the ratio per discs is not known so far, the discrete concentrations of scaffold and membrane protein could not be determined. Total protein concentrations for all measurements were in between 1.5 mg/ml and 5 mg/ml. Only in the case of PR, the concentration of the inserted membrane protein could be determined by the specific absorbance of green-light absorbing PR at 530 nm with an extinction coefficient of 44,000 $M^{-1}cm^{-1}$.

## SDS-PAGE

Protein samples were prepared by addition of SDS loading buffer (100 mM Tris-HCl pH 6.8, 8 M Urea, 0.12% (w/v) bromphenoleblue, 20% (w/v) SDS, 15% (v/v) glycerol, 20% (v/v) $\beta$-mercaptoethanol) and separated by Tris-Tricine SDS-PAGE (*Schägger, 2006*). Therefore, 11% Tris-Tricine gels were run for 15 min at 80 V and for 45–55 min at 150 V. For protein staining, gels were incubated for 20 min with a solution of 50% (v/v) ethanol and 10% (v/v) acetic acid, washed with 200 ml of ddH$_2$O and incubated with a colloidal staining solution (0.02% (w/v) Commassie Brilliant Blue G250, 5% (w/v) aluminium sulphate-(14-18)-hydrate, 10% (v/v) ethanol and 2% (v/v) ortho-phosphoric acid).

## Size-exclusion chromatography

Analytical size-exclusion chromatography was performed on an Äkta Purifier FPLC system using a Superdex 200 3.2/30 or a Superdex 200 3.2/30 Increase column as described elsewhere (*Roos et al., 2012*) with SEC-buffer (40 mM Tris-HCl pH 8.0, 100 mM NaCl) and a flow rate of 0.05 or 0.075 ml/min, respectively.

## LILBID-MS

Nanodisc-samples were buffer exchanged into 50 mM ammonium acetate at pH 6.8 directly before measurement with LILBID-MS. Zeba Micro Spin Desalting Columns (article number 89887) from Thermo Scientific were used to exchange the buffer, operating with a 7 kDa cut-off filter. To equilibrate the desalting column, it was washed five times with the buffer at 1500 rpm for 1 min and the final sample buffer exchange was done for 2 min at 1500 rpm. Sample concentration was within a range between 10 and 100 µM. 3 µL of buffer exchanged sample was used for MS.

A piezo-driven droplet generator (MD-K-130 from Microdrop Technologies GmbH, Norderstedt, Germany) was used to produce droplets of 30 µm diameter with a frequency of 10 Hz at 100 mbar. The droplets are transferred to high vacuum and irradiated by an IR laser operating at 2.94 µm, a vibrational absorption wavelength of water. The laser is a pulsed Nd:YAG laser, pumping an optical parametric oscillator, converting the wavelength to 2.94 µm. Laser pulse length is 6 ns, and the pulse

energy output is between 9.5 and 23 mJ. LILBID settings have previously been published in detail (*Morgner et al., 2006*). Laser irradiation leads to an explosive expansion of the sample droplet and ions are released which were accelerated by a pulsed electric field and analyzed by a homebuilt reflectron-time-of-flight (TOF) setup. Ion acceleration is based on a Wiley-McLaren type ion source with a delayed acceleration pulse of 370 µs and an overall acceleration of 6.6 kV, working at $10^{-5}$ mbar.

Data detection and processing was done with the software *Massign* (*Morgner and Robinson, 2012*) available at http://massign.chem.ox.ac.uk/. MS spectra show averaged signals between 1000 and 1500 droplets. The spectra were normalized with the software *OriginPro 2016*.

To calculate the number of attached lipids on the scaffold protein from empty nanodiscs, the full peak width at half maximum height (FWHM) of the scaffold protein was subtracted from the FWHM monomeric scaffold protein attached with lipids and divided by the lipid mass. For the statistical analysis of the distribution of different species within the spectra of PR, the integrals of the single peaks were normalized to the overall peak area. Mean values and standard deviation of 2–3 measurements were calculated. This was done separately for every of the three spectra with different laser intensities.

## Acknowledgements

We thank Simone Prinz and Prof. Werner Kühlbrandt from the MPI of Biophysics, Frankfurt am Main for taking EM images of empty nanodiscs. We as well thank the P4EU network for valuable support. EH and OP are supported by Collaborative Research Center (SFB) 807 of the German Research Foundation (DFG) FB received funding from Instruct, part of the European Strategy Forum on Research Infrastructures (ESFRI). AL is funded by the German Research Foundation (DO545/11) BH was supported by NIH (U54GM087519) and by the International Max Planck Research School for Structure and Function of Biological Membranes VD and NM are supported by Cluster of Excellence Frankfurt (Macromolecular Complexes) NM received funding from the European Research Council under the European Union's Seventh Framework Programme (FP7/2007–2013) / ERC Grant agreement n° 337567.

## Additional information

### Competing interests

VD: Reviewing editor, eLife. The other authors declare that no competing interests exist.

### Funding

| Funder | Grant reference number | Author |
| --- | --- | --- |
| Deutsche Forschungsgemeinschaft | Collaborative Research Center (SFB) 807 | Erik Henrich Oliver Peetz Christopher Hein |
| Deutsche Forschungsgemeinschaft | DO545/11 | Aisha Laguerre |
| National Institutes of Health | U54GM087519 | Beate Hoffmann |
| Max Planck Research School for Structure and Function of Biological Membranes | | Beate Hoffmann |
| Deutsche Forschungsgemeinschaft | Cluster of Excellence Frankfurt | Volker Dötsch Nina Morgner |
| European Strategy Forum on Research Infrastructures | Instruct | Frank Bernhard |
| P4EU | | Frank Bernhard |
| European Research Council | European Union's Seventh Framework Programme (FP7/2007-2013)/ ERC Grant agreement n° 337 | Nina Morgner |

The funders had no role in study design, data collection and interpretation, or the decision to submit the work for publication.

## Author contributions
EH, JH, FB, NM, Conceptualization, Software, Formal analysis, Supervision, Funding acquisition, Validation, Investigation, Visualization, Methodology, Writing—original draft, Writing—review and editing; OP, CH, AL, Conceptualization, Formal analysis, Investigation, Visualization, Writing—original draft, Writing—review and editing; BH, Conceptualization, Investigation, Methodology, Writing—original draft, Writing—review and editing; VD, Investigation, Methodology, Writing—original draft, Writing—review and editing

## Author ORCIDs
Volker Dötsch, http://orcid.org/0000-0001-5720-212X
Nina Morgner, http://orcid.org/0000-0002-1872-490X

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
