## [Decision Letter]

Thank you for submitting your article "Analyzing native membrane protein assembly in nanodiscs by combined non-covalent mass spectrometry and synthetic biology" for consideration by *eLife*. Your article has been favorably evaluated by Richard Aldrich (Senior Editor) and three reviewers, one of whom is a member of our Board of Reviewing Editors. The reviewers have opted to remain anonymous.

The reviewers have discussed the reviews with one another and the Reviewing Editor has drafted this decision to help you prepare a revised submission.

Summary:

Morgner et al. present novel and interesting data on the use of LILBID MS to investigate transmembrane protein (complexes) in nanodiscs. There is a considerable number of different proteins, lipids and scaffold proteins shown and a relatively clear picture of the performance of "AMPLE-MS" for the characterization of these reconstituted particles emerges. The reviewers were positive in their evaluation of the work, but significant concerns were voiced.

The key shared concern was the use of LILBID MS as a bundled "novel" approach with protein production into nanodiscs. The production part has been described elsewhere and is not novel. Furthermore, it is unclear whether in vitro protein production in the context of LILBID MS provides any advantages over other ways of producing nanodisc-incorporated protein samples. Introduction of a new term "AMPLE-MS" seems unnecessary. Additionally, the experimental description is insufficient. There should be a much more detailed description of how the experiments are performed and analyzed. More specific concerns are:

Essential revisions:

1) While results are significant and their novelty warrants publication, the key findings are somewhat misrepresented. In particular, the use of cell-free expression (which facilitates direct transfer of lipid-bound protein into nanodiscs) is mixed with the novel MS approach (at least for this type of application) in a way, which may confuse the reader and blur the line between what is novel and what is not. Cell-free expression has been described before, and so has the production of lipid nanodiscs. Protein expression and purification in detergent is recently used to form protein-lipid nanodiscs, so how does the use of LILBID MS rely on cell-free expression? What is its performance with lipid nanodiscs, which were prepared differently, i.e. the "classic" way? How does LILBID compare with nano-ESI (see literature)? The innovation presented in this paper lies in the use of LILBID, but it is unclear how this is linked with the alternative origins and preparation methods of lipid nanodiscs. If the authors believe this to be the case, they should show the evidence.

2) If the nanodisc-based production were retained in the manuscript, then more details would be required. How does one select lipoproteins for the formation of the nanodiscs? Are there experimental advantages/disadvantages of distinct proteins in view of the proposed methodologies? What are checkpoints in the process; i.e. how does one know that the proteins have been properly expressed and folded in the nanodiscs? In general, the description of the cell-free expression into nanodiscs is only cursory.

3) How would one determine whether the observed species reflect heterogeneity of the sample following cell-free expression or disruption of the complex during gas phase formation? Could multiple copies of monomeric proteins be inserted into the same nanondiscs in the absence of protein:protein interactions? How would one know? Finally, it is unclear form the manuscript how common is LILBID instrumentation setup. If it is not common, how easy is it to establish based on more commonly accessible instruments. What are the advantages (current or potential) of LILBID compared to more common ESI?

4) The term "synthetic biology" is uncommon for cell-free expression. Statements are also made about the use of detergents (e.g. bacteriorhodopsin), which while true are somewhat misleading. In order to natively reconstitute a transmembrane protein, a suitable environment needs to be provided, i.e. the *correct* lipid or lipid mixture, or the *correct* detergent (which is obviously meant to mimic the lipid). Just comparing some lipids with some detergents is not sufficient to conclude which method is better.

5) The laser power required to resolve individual masses of nanodiscs is too high to keep them intact. All the investigations described in Figure 2 and supplements seem to be of the remaining few lipid molecules, not reflecting intact nanodiscs. It is not explained in sufficient detail what is the value of these experiments. The use of shorter lipids is based on these controls, but it is not clear how critical is this selection.

6) In all experiments, attention is given to resolving partially dissociated species. How is that information used to infer biologically relevant conclusions?

7) Subsection “Oligomeric state analysis of membrane proteins in nanodiscs”, second paragraph: if that was true, then switching to diff. scaffold proteins or isotopes should show such species, but I don't think that they are actually released intact from the nanodiscs! This is too important to brush over with a "cannot yet be assigned" comment. Please elaborate this point in the paper.

8) Subsection “Refinement of LILBID-MS signal resolution by isotope labeling”, first paragraph: what type of deuterium labeling is used (during expression or later?) and how stable do the authors think that such labels will be?

9) A reviewer is not fully convinced by data on MraY from *B. subtilis* and *E. coli*. It seems clear that there is dimer of *B. subtilis* MraY in both DMPG and DMPC, but the amount of a monomer/nanodisc complex is much greater in DMPC. In contrast, the amount of dimer is very small, barely out of the noise. This does not correlate with comparable activity observed in these two lipids. Overall, it is not clear whether similar amounts of proteins were produced and analyzed. SDS gels showing the protein assembled into different lipids would have been helpful. In contrast, inclusion of previously published data in the figure seems unnecessary.

10) The conclusion needs to be rewritten as it mixes and confuses issues of protein production, nanodisc preparation and MS characterization.

11) Discussion, third paragraph and elsewhere: I disagree that the spectra are straightforward to analyze. The authors use specialized, home-written software and peak shapes are often ambiguous (see above). More importantly, while I agree that the largest complex is here always matching the expected species, there are plenty of smaller subcomplexes visible. Are they due to fragmentation? What mechanism (asymmetric CID?) Often the intact protein complex cannot be released from the nanodisc or only forms a minor species among many in the spectra.

---

## [Author Response]

*Summary:*

*Morgner et al. present novel and interesting data on the use of LILBID MS to investigate transmembrane protein (complexes) in nanodiscs. There is a considerable number of different proteins, lipids and scaffold proteins shown and a relatively clear picture of the performance of "AMPLE-MS" for the characterization of these reconstituted particles emerges. The reviewers were positive in their evaluation of the work, but significant concerns were voiced.*

*The key shared concern was the use of LILBID MS as a bundled "novel" approach with protein production into nanodiscs. The production part has been described elsewhere and is not novel. Furthermore, it is unclear whether* in vitro *protein production in the context of LILBID MS provides any advantages over other ways of producing nanodisc-incorporated protein samples. Introduction of a new term "AMPLE-MS" seems unnecessary. Additionally, the experimental description is insufficient. There should be a much more detailed description of how the experiments are performed and analyzed. More specific concerns are:*

We follow the recommendation of the reviewers and omit the term “AMPLE-MS”, while presenting a more detailed and structured description of our strategy. In particular, the advantages of extraction-and detergent-free preparation of membrane protein/nanodisc samples by cell-free expression are better indicated and discussed. We agree that the novel aspect of the work is rather the combination of the two previously described techniques, resulting into considerable synergies.

Overall, the Introduction and Discussion part of the manuscript have been completely re-written and major modifications have been made in the Results section. Furthermore, additional experiments have been done and two new figures have been added.

*Essential revisions:*

*1) While results are significant and their novelty warrants publication, the key findings are somewhat misrepresented. In particular, the use of cell-free expression (which facilitates direct transfer of lipid-bound protein into nanodiscs) is mixed with the novel MS approach (at least for this type of application) in a way, which may confuse the reader and blur the line between what is novel and what is not. Cell-free expression has been described before, and so has the production of lipid nanodiscs. Protein expression and purification in detergent is recently used to form protein-lipid nanodiscs, so how does the use of LILBID MS rely on cell-free expression? What is its performance with lipid nanodiscs, which were prepared differently, i.e. the "classic" way? How does LILBID compare with nano-ESI (see literature)? The innovation presented in this paper lies in the use of LILBID, but it is unclear how this is linked with the alternative origins and preparation methods of lipid nanodiscs. If the authors believe this to be the case, they should show the evidence.*

LILBID-MS does not principally depend on cell-free expression and it can be postulated that membrane protein/nanodisc complexes prepared by the “classical” post-translational reconstitution process involving detergent solubilization could also be analyzed. Previous reports showed the ESI-MS analysis of comparable samples prepared by the classical post-translational mechanism and corresponding references have been cited (see Introduction, third paragraph). However, the stoichiometry of membrane protein assemblies could be affected by the rigid membrane extraction and detergent solubilization process used for the “classical” strategy. One discussed example is proteorhodopsin, which is monomeric in the particular detergent DH7PC but multimeric in lipid environments (see new Figure 6—figure supplement 1). Furthermore, many membrane proteins are hard to synthesize in traditional cell-based expression systems or denature upon membrane extraction or contact with detergents. One shown example is *E. coli* MraY, which was completely inactive and aggregated in presence of detergents. This sample cannot be prepared by the classical technique. Therefore, cell-free expression largely extends the portfolio of membrane protein complexes that can be analyzed by LILBID-MS or similar techniques, it removes uncertainties and risks due to the necessary detergent treatment with classical protocols and it significantly accelerates and streamlines the overall sample preparation process concomitant with enabling numerous labelling and sample refinement opportunities (see Introduction, third paragraph).

We have rewritten the Introduction and Discussion sections in order to make these basic considerations and resulting advantages of our strategy clearer.

*2) If the nanodisc-based production were retained in the manuscript, then more details would be required. How does one select lipoproteins for the formation of the nanodiscs? Are there experimental advantages/disadvantages of distinct proteins in view of the proposed methodologies? What are checkpoints in the process; i.e. how does one know that the proteins have been properly expressed and folded in the nanodiscs? In general, the description of the cell-free expression into nanodiscs is only cursory.*

The in vitro production of empty nanodiscs and the co-translational integration of membrane proteins by cell-free expression was described in detail before and we have cited the references. For a better description the Methods section has been expanded (see subsections “Nanodisc preparation” and “Cell-free protein production”). Key advantages are the detergent-free process along with other general advantages of cell-free expression such as efficient labelling (see Introduction, first paragraph and following). We have now highlighted this better in the manuscript. In addition, a more detailed overview of the nanodisc production technique also indicating critical checkpoints, is included (see subsection “Nanodisc preparation”). Proper expression of all proteins could be revealed by MS-spectra or e.g. SDS-PAGE. Within this study, we only used proteins where the functional characterization of the corresponding samples has already been published in previous papers which have now been cited.

*3) How would one determine whether the observed species reflect heterogeneity of the sample following cell-free expression or disruption of the complex during gas phase formation? Could multiple copies of monomeric proteins be inserted into the same nanondiscs in the absence of protein:protein interactions? How would one know?*

This is an important consideration. As we learned from measurements of different protein complexes, we can monitor their stoichiometry present in nanodiscs. The nanodisc complexes may differ in the amount of proteins they carry, but always have two scaffold proteins. A key point of our platform is that we can probe the tightness of protein interactions by tuning the laser power of LILBID-MS (see subsection “Oligomer formation of proteorhodopsin (PR) in nanodiscs”, second paragraph). So, weak non-specific interactions, e.g. in between membrane proteins and the scaffold proteins or in between two different membrane protein complexes inside one nanodisc, will dissociate first at already low laser power. “Real” membrane protein complexes having a tight and specific interface will only disrupt at higher laser power. Tuning the disruption of the membrane protein/nanodisc complexes leads then to gradually loss of integrated membrane proteins as well as of scaffold proteins (see new Figure 6). Spectra taken at lower laser power still reveal the integrated membrane proteins together with both scaffold proteins and thus could indicate potential sample heterogeneity, e.g. by multiple integration of monomeric or dimeric proteins. However, spectra taken at higher laser power, and this was the routine measurement condition in the manuscript, show the dissociation of the sample into complexes still remaining one or two scaffold proteins, but also gave signals of the free membrane proteins. The maximum stoichiometry of these free protein complexes, which have already lost the non-specific interaction to both scaffold proteins, therewith reflect the highest oligomeric stoichiometry of the particular membrane protein (see the aforementioned paragraph). We speculate, and this is supported by the results obtained with all six analyzed membrane proteins, that potential non-specific interaction in between multiple inserted proteins within one nanodisc membrane would not stay together after applying laser power conditions that already result into loss of both scaffold proteins.

As a note, we have not observed evidence of such a multiple insertion even with the analyzed relatively small monomeric or dimeric membrane proteins (see Figure 4 and Figure 5). However, multiple integration may also depend on experimental conditions such as nanodisc size and concentration. So we would propose that it may occur, but with our strategy we either won’t detect it or we could easily identify it.

In short: Heterogeneity in complexes with two scaffold proteins reflect a heterogeneity of complexes present in solution. The highest observed Oligomeric state of a protein complex without scaffold proteins determine the biologically relevant stable complex.

We added an additional Figure (Figure 6) showing a representative tuning of complex dissociation by applying different laser powers, and we added a better explanation of the theory behind (subsection “Oligomer formation of proteorhodopsin (PR) in nanodiscs”, third paragraph).

*Finally, it is unclear form the manuscript how common is LILBID instrumentation setup. If it is not common, how easy is it to establish based on more commonly accessible instruments. What are the advantages (current or potential) of LILBID compared to more common ESI?*

LILBID-MS is an emerging technique and we hope to increase its application by presenting results such as shown in this manuscript. The major difference if compared with Esi-MS is the mechanism of gas-phase transition of the analyzed sample. We use a single power transfer of a 6 nsec laser pulse while Esi-MS is based on a permanent ion collision mechanism. These differences could have significant effects on sample stability and dissociation kinetics. A corresponding study that directly compares Esi-MS with LILBID-MS by using identical samples is currently in preparation.

We added a sentence specifying the difference between the two MS techniques into the Introduction.

*4) The term "synthetic biology" is uncommon for cell-free expression.*

The term has recently been introduced and cell-free expression is accepted as a core platform in synthetic biology. We have added corresponding references (Hodgman & Jewett, Metab Engin 14, 261-269, 2013, or Smith et al., FEBS Lett 2014, 588: 2755-61.) (See Introduction, third paragraph).

*Statements are also made about the use of detergents (e.g. bacteriorhodopsin), which while true are somewhat misleading. In order to natively reconstitute a transmembrane protein, a suitable environment needs to be provided, i.e. the correct lipid or lipid mixture, or the correct detergent (which is obviously meant to mimic the lipid). Just comparing some lipids with some detergents is not sufficient to conclude which method is better.*

We completely agree that many membrane proteins stay folded and even in oligomeric assemblies in detergent micelles. However, to find out which detergent or detergent combination is suitable for a given membrane protein can become a time-consuming task. Membrane extraction and detergent treatment is a relatively harsh procedure and represents always a risk of complex disintegration. Whether oligomeric complexes re-assemble upon the classical reconstitution procedure into membranes cannot be predicted. Our strategy eliminates the solubilization step and the proteins are directly analyzed in the initial membrane, in which they have been co-translationally inserted (see Introduction, first paragraph and Figure 1). We therefore propose that our strategy provides a less risky procedure to view the native situation of membrane protein complexes and we support this by the analysis of six different systems (see Introduction, last paragraph).

In the discussed example of proteorhodopsin, also the monomeric form in the detergent DH7PC folds into its active conformation and its NMR structure has been solved in that detergent. In the detergent DDM it forms multimeric complexes similar to that in native membranes and we have included a corresponding reference of that (see subsection “Oligomer formation of proteorhodopsin (PR) in nanodiscs”, last paragraph). However, this example shows that functional assays alone must not be a proof of native conformation (see also Laguerre et al. 2016, which has been included now). The direct co-translational insertion of proteorhodopsin into nanodisc membranes results into the native multimeric complexes and eliminates tedious screening of detergents supporting multimer formation by photocycle kinetics. With the second example of *E. coli* MraY, we have previously shown that contact with detergent completely inactivates the protein (see corresponding citation Ma et al. 2011). The established strategy of detergent-free production with nanodiscs is therefore the only way to isolate this protein for in vitro characterization.

*5) The laser power required to resolve individual masses of nanodiscs is too high to keep them intact. All the investigations described in Figure 2 and supplements seem to be of the remaining few lipid molecules, not reflecting intact nanodiscs. It is not explained in sufficient detail what is the value of these experiments. The use of shorter lipids is based on these controls, but it is not clear how critical is this selection.*

These have been basic experiments in order to define optimal measurement conditions and to improve resolution. Figure 2 shows the different “stickiness” of the individual lipids on the scaffold proteins. This correlates with the “stickiness” to other proteins as well. Therefor the lipids which are less “sticky” are chosen for the rest of the experiments, as they allow for better spectral resolution (see subsection “LILBID-MS analysis of empty nanodiscs”, last paragraph). Depending of the mass differences one wants to observe, the lipid selection is more or less critical. An explanatory sentence is added to the aforementioned subsection and a new figure (see new Figure 6—figure supplement 1) has been added for better illustration. Further refinement of the LILBID-MS technique will allow to stepwise implement a higher variety of different lipid types.

*6) In all experiments, attention is given to resolving partially dissociated species. How is that information used to infer biologically relevant conclusions?*

Please also refer to answer to point #3. The laser dissociation of the assemblies depends on the applied laser power in the experiment and on the strength of interaction. The largest complex in the spectra thus represents the highest stably incorporated oligomerization state and the biologically relevant complex in the membrane. The partially dissociated species under higher laser conditions originate from dissociation and appear with no or one scaffold protein as well. In order to support this assumption, we have analyzed several well-studied model proteins having different oligomeric forms such as proteorhodopsin (hexamer), KcsA (tetramer), EmrE (dimer) and LspA (monomer). In all cases, the detected highest oligomeric form exactly matches with the expected complex. Therefore, we think that our strategy is indeed suitable to analyze the biological relevant complexes in its native stoichiometry. However, for the dissociated lower order complexes, it cannot be ruled out that some of them may already exist in addition within the membrane as well. These would be observable as lower smaller complex with two scaffold proteins under low laser conditions. This should be clearer now from the changes we made in the second paragraph of the subsection “Oligomer formation of proteorhodopsin (PR) in nanodiscs” and Figure 6, and we further modified the statement on the interpretation of the spectra in the Discussion accordingly.

*7) Subsection “Oligomeric state analysis of membrane proteins in nanodiscs”, second paragraph: if that was true, then switching to diff. scaffold proteins or isotopes should show such species, but I don't think that they are actually released intact from the nanodiscs! This is too important to brush over with a "cannot yet be assigned" comment. Please elaborate this point in the paper.*

As seen later in the manuscript, the proposed species could be released intact from the nanodiscs. This issue is picked up in second paragraph of the subsection “Oligomeric state analysis of membrane proteins in nanodiscs”, where we show that labelling now allows for the unambiguous assignment.

*8) Subsection “Refinement of LILBID-MS signal resolution by isotope labeling”, first paragraph: what type of deuterium labeling is used (during expression or later?) and how stable do the authors think that such labels will be?*

The labels are applied during expression by the addition of labeled amino acids (see subsection “Cell-free protein production”). Besides some minor amino acid scrambling in the cell-free lysate specific to asparagine and glutamine residues, this labelling is extremely stable even at high temperatures used for NMR measurements. The labelling strategies by cell-free expression have been published numerous times before and they have been used to facilitate full assignments of rather large membrane proteins within NMR. We have added some corresponding references (Laguerre 2015,2016 and Reckel 2010).

*9) A reviewer is not fully convinced by data on MraY from B. subtilis and E. coli. It seems clear that there is dimer of B. subtilis MraY in both DMPG and DMPC, but the amount of a monomer/nanodisc complex is much greater in DMPC. In contrast, the amount of dimer is very small, barely out of the noise. This does not correlate with comparable activity observed in these two lipids. Overall, it is not clear whether similar amounts of proteins were produced and analyzed. SDS gels showing the protein assembled into different lipids would have been helpful. In contrast, inclusion of previously published data in the figure seems unnecessary.*

We apologize for that confusion; we zoomed in one spectrum for better illustration but forgot to indicate that (see revised Figure 7). Both spectra are now shown at identical scale and dimer formation in both lipids is quite comparable. The larger peak area of the monomer/scaffold protein signal with DMPC represents a dissociation product resulting from non-specific interactions in between MraY and the scaffold proteins. Those might be influenced by the different charge of DMPC versus DMPG, but are not of relevance for the interpretation of our results. Samples were always purified by Strep-tags of the inserted membrane proteins (see subsection “Purification of cell-free expressed membrane proteins”, first paragraph) and analyzed at similar concentrations of approx. 1.5 mg/mL. Yes, we agree that showing already published data is not necessary but we think that here it illustrates the results nicely and makes it simple for the reader to follow without having to download and going deeply into previous publications.

*10) The conclusion needs to be rewritten as it mixes and confuses issues of protein production, nanodisc preparation and MS characterization.*

The Discussion has been completely re-organized and re-written.

*11) Discussion, third paragraph and elsewhere: I disagree that the spectra are straightforward to analyze. The authors use specialized, home-written software and peak shapes are often ambiguous (see above).*

What is meant here is that the spectra are comparably easy to interpret (e.g. if compared with ESI spectra), as the LILBID desorption process leads to only few, very low charged gas phase ions (see Discussion, second paragraph). In the case of the here shown species, mainly the singly charged ones are observed. While the home-written software is helpful, it is not essential. However, the software is open accessible and the corresponding URL has been included. This is clarified now in the third paragraph of the subsection “LILBID-MS”.

*More importantly, while I agree that the largest complex is here always matching the expected species, there are plenty of smaller subcomplexes visible. Are they due to fragmentation? What mechanism (asymmetric CID?) Often the intact protein complex cannot be released from the nanodisc or only forms a minor species among many in the spectra.*

See also comments to point #3 and #6.

The intact complex can be released in all cases, if sufficient laser energy is used. This leads at the same time to fragmentation of part of these complexes and therewith to the detection of the full-size complex along with potentially all possible lower size complex fragments. This mechanism should now become clearer from the changes made in the second paragraph of the subsection “Oligomer formation of proteorhodopsin (PR) in nanodiscs”.